# Factors and Mechanisms of Thyroid Hormone Activity in the Brain: Possible Role in Recovery and Protection

**DOI:** 10.3390/biom14020198

**Published:** 2024-02-07

**Authors:** Laura Sabatino, Dominga Lapi, Cristina Del Seppia

**Affiliations:** 1Institute of Clinical Physiology, National Council of Research, Via Moruzzi 1, 56124 Pisa, Italy; cristina.delseppia@cnr.it; 2Department of Biology, University of Pisa, 56127 Pisa, Italy; dominga.lapi@unipi.it

**Keywords:** thyroid hormones, TH deiodinases, TH transporters, brain damage, cognitive impairment

## Abstract

Thyroid hormones (THs) are essential in normal brain development, and cognitive and emotional functions. THs act through a cascade of events including uptake by the target cells by specific cell membrane transporters, activation or inactivation by deiodinase enzymes, and interaction with nuclear thyroid hormone receptors. Several thyroid responsive genes have been described in the developing and in the adult brain and many studies have demonstrated a systemic or local reduction in TH availability in neurologic disease and after brain injury. In this review, the main factors and mechanisms associated with the THs in the normal and damaged brain will be evaluated in different regions and cellular contexts. Furthermore, the most common animal models used to study the role of THs in brain damage and cognitive impairment will be described and the use of THs as a potential recovery strategy from neuropathological conditions will be evaluated. Finally, particular attention will be given to the link observed between TH alterations and increased risk of Alzheimer’s Disease (AD), the most prevalent neurodegenerative and dementing condition worldwide.

## 1. Introduction

THs regulate differentiation, growth, and energy metabolism in virtually all cells and tissues in all vertebrates, by affecting the expression of different sets of genes. In the brain, THs are essential for correcting brain maturation. They influence neurogenesis, neuronal and glial cell differentiation and migration, synaptogenesis, and myelination. Thyroid hormone deficiency may severely affect the brain during fetal and postnatal development, causing retarded maturation, intellectual deficits, and neurological impairment [1,2,3,4].

Thyroxine (T4) is considered a prohormone which is converted to the active form, triiodothyronine (T3). THs are synthesized in the thyroid in large part in the form of T4 (in a ratio of approximately 14:1 with respect to T3), are released in the circulation mostly bound to transport proteins, and reach the target tissues where specific transporters mediate the uptake of TH by the cells [5,6,7,8].

At the central level, TH homeostasis is regulated by the hypothalamic–pituitary–thyroid (HPT) axis through the activity of TSH-releasing hormone (TRH), Thyroid Stimulating Hormone (TSH), and the negative feedback on the HPT axis by circulating T3 and T4. Conversely, at the cellular level, T3 and T4 undergo to a strict homeostatic control relying on integrated transporters activity mediating the uptake of TH inside the cell, and on the three iodothyronine deiodinase enzymes, D1, D2, and D3, which regulate the activation and inactivation of TH and their metabolites [9,10] (Figure 1).

At the cell level, THs mediate both genomic and nongenomic effects [11,12]. TH action is generally defined as “genomic” when it directly involves the interaction of the biologically active T3 with specific nuclear thyroid hormone receptors (TRs), and “nongenomic” when TH action does not involve nuclear TRs but requires the interaction of T4 with plasma membrane integrin αvβ3 receptor, which activates specific intracellular signaling pathways [13,14].

The aim of this review is to discuss the principal factors and mechanisms regulating TH effects in cells, with a special focus on the role of TH signaling in brain disease and injury and on the protective effects induced by TH administration in experimental models. Moreover, THs’ protective role in cognitive/emotional impairment will be also evaluated.

## 2. THs and Normal Brain Development

THs are essential for normal brain development as they regulate the differentiation, migration, maturation, signaling, and metabolism of neurons. The function of THs is to provide the right temporal signal for the different stages of brain development to proceed, and the integration of different neuronal systems and association of glia cells to occur [15]. An incorrect synchronism in such developmental processes would inevitably lead to permanent deleterious outcomes. TH action is particularly critical starting from the late fetal stages until 3–4 weeks of the postnatal period in rodents [16], and the first postnatal month in humans [3]. For these reasons, most research on the role of THs in brain development has concentrated on the perinatal and early postnatal period and both the absence or the excess of THs in this critical time window can cause significant and irreversible structural and functional damage, with a relevant impact on the efficiency of the neurotransmitter system [17]. The presence of TH signaling components at very early stages of brain development, before the onset of fetal thyroid function, points to the maternal THs supply as a determining factor for proper fetal brain development in the early stages [18]. Interestingly, human data indicate that the maternal contribution to the fetus, even though in a smaller percentage, fulfils an important protective role until birth [3]. In fact, children born to mothers with thyroid disease experience an increased risk of neurologic and psychiatric diseases later in life [19,20]. Many of these problems are the expression of impaired TH signaling at different component levels, such as TH transporters, deiodinases, and TRs, but also reflect the alteration in the development of neurotransmitters in the central nervous system [21].

## 3. Nuclear TH Receptors

The genomic action of THs requires the mediation of nuclear TRs, which are ligand-activated transcription factors directly interacting with thyroid responding elements (TREs) present in the promoters of target genes and regulating their transcription [11]. Three main hormone-binding TRs are encoded by TRα and TRβ genes (THRA and THRB for human genes, Thra and Thrb for mice/rat genes) by alternative splicing (TRα1, TRβ1, and TRβ2) and show different affinities for the TRE sequences, but also for proteins and co-factors involved in gene transcription regulation [22]. The three main isoforms of TRs are expressed at variable levels in the brain. TRα1 is the predominant subtype; it is almost ubiquitous and is expressed since the earliest stages of development [23], while TRβ1 is the least expressed in the brain, and TRβ2, expressed in the hypothalamus, pituitary, cochlea, and retina, is considered responsible for T3-dependent negative feedback for TRH hypothalamic production and, consequently, for TSH release by the pituitary [24,25]. TRs account for many important functions of THs in embryonic and adult life and can interact with DNA both in the presence (activation) or absence (repression) of ligands [22,26]. Furthermore, non-T3 binding TRα isoforms have been described (TRα1 and TRα2) in the fetal, neonatal, and adult brain. Unliganded receptors (aporeceptors) are not silent but may have an opposite effect with respect to liganded receptors (holoreceptors); therefore, a T3-positively regulated target gene may be likely repressed by the aporeceptor and vice versa [26]. This is particularly relevant in the developing brain, where the control of T3 levels must be carefully calibrated and unliganded receptors may have a critical regulatory role in neuronal differentiation [27].

## 4. TH Transporters

Free THs enter the cells through transmembrane transporter proteins [28]. Many different carriers, belonging to several families, have been described to mediate TH transport across the plasma membrane [8,29]. Among them, the monocarboxylate transporter 8 (MCT8) is considered very specific for THs, with a higher affinity for T3 than T4 [30,31]; it is expressed in many tissues and organs and is determinant in the increase in TH intracellular availability [31,32]. Furthermore, MCT8 is also important for the transport of inactive metabolites such as reverse T3 (rT3) and 3,3′-diiodothyronine (T2) [31]. A mutation in the gene for MCT8 (SLC16A2) results in the impairment of T3 uptake in the neurons, leading to the neurological deficit known as Allan–Herndon–Dudley syndrome (AHDS), an X-linked disorder, characterized by hypotonia, spasticity, muscle weakness, neurological disorders, and cognitive impairment [33,34,35]. AHDS patients have high levels of T3, with borderline low T4 and high/normal TSH, with symptoms of hyperthyroidism in the peripheral tissues and of hypothyroidism at the central nervous system level. Moreover, the expression of MCT8 at the blood–brain barrier (BBB) level also suggests the role of this transporter in the T3 uptake by BBB [36]. Mouse models are commonly used to study intracellular TH signaling; however, although the expression pattern of MCT8 in the mice is similar to that of humans, Mct8-deficient mice do not show a neurological phenotype analogous to humans but only the same endocrine alterations observed in humans. Interestingly, in the brain of mice deprived of Mct8 function, the uptake of exogenous T3 is severely reduced, whereas T4 uptake is reduced by 50% with respect to the wild type [37]. These findings suggest that additional TH transporters, called “secondary TH transporters”, might be involved to compensate for Mct8 deprivation in mice [36,37]. Primary TH transporters are extremely specific for TH uptake by the target cells, whereas the secondary TH transporters can also mediate the uptake into the cells of various kinds of compounds, other than THs, and include the transporter of aromatic amino acids (MCT10), organic anion transporting polypeptides (OATPs), and the large neutral amino acid transporters (LAT1 and LAT2) [30,38,39]. OATP2B1, OATP3A1, and OATP4A1 are expressed ubiquitously, whereas other members of the OATP family, including OATP1B1, OATP1B3, and OATP1C1, have a more restricted expression, the latter being predominantly localized in the capillary endothelium and choroid plexus [40]. Furthermore, OATP1C1 is considered the principal responsible for T4 uptake from the circulation to the brain across the BBB, where it is converted to T3 and then transported into neurons by MCT8 [41,42]. At the brain level, LAT1 and LAT2 are expressed in the luminal and abluminal membranes of brain capillary endothelial cells of the BBB, and LAT1 is considered the most active isoform [43]. The system of TH transporters is very interesting from an evolutionary point of view since it shows how structurally different protein families can converge to the same common function (Figure 2).

At the moment, no effective therapy is available to prevent or treat AHDS; however, several efforts have been made to find some TR-activating compounds which can cross the BBB and enter the target cells, independently from MCT8. An efficient treatment of this syndrome should be able to correct the hypothyroidism state in the central nervous system and the hypermetabolic condition due to an excess of T3 at the peripheral tissues. The first tested thyromimetic compound was 3,5-diiodothyropropionic acid (DITPA), which, once administered to Mct8-knockout (KO) mice, enters the liver and brain and is able to restore the correct TH signaling in the cells, reducing the excessive expression of T3-responsive genes and regulating D1 and D2 activities in a dose-dependent manner [44,45]. Other compounds studied in the context of AHDS are 3,3′,5-triiodothyroacetic acid (Triac), and 3,3′,5,5′-tetraiodothyroacetic acid (Tetrac), two naturally occurring metabolites of THs, which are less concentrated and with shorter half-lives than T4 and T3 [46]. As the other analogues, Triac is transported into the brain cells by a transporter different from MCT8 and is efficiently metabolized by D1 and D3 and affects the expression of TH-responsive genes in neurons [47]. However, Mct8-KO mice do not reproduce the neurological phenotype observed in AHDS patients, which makes this model not adequately reliable for what is observed in humans.

To overcome this experimental limit, an Mct8/Oatp1c1 double KO (DKO) mouse model was developed, in which the compensating mechanism of T4 to T3 conversion, mediated by D2 and observed in Mct8-KO mice, is also disrupted [48,49,50]. These DKO mice show both peripheral TH homeostasis and neurocognitive phenotype impairment and are therefore considered the most representative mouse model of AHDS [48,49]. In particular, DKO mice showed altered Purkinje cell dendritogenesis, with compromised cerebellar development and myelination, altered deiodinase activities, and TH target gene expression [48,49].

To date, complete phenotypical and neurological data on MCT8 deficiency are still lacking, due to the different approaches used in different studies; therefore, important pathophysiological aspects remain undefined and treatment options are limited. In a recent trial on pediatric and adult patients with MCT8 deficiency, some improving effects were reported after Triac treatment on peripheral thyrotoxicosis and neurocognitive outcomes [51]. Further studies on long-term Triac treatment in real-life settings confirmed the beneficial effects of Triac on the peripheral phenotype of MCT8-deficient patients and its clinical potential in AHDS [52].

## 5. TH Deiodinases

TH deiodinases are crucial in the functional diversification of TH signaling and are main actors in the fine regulation of TH homeostasis. The three deiodinases are synthesized by three different genes, which have high sequence similarity. The active site of the three enzymes is highly conserved and contains the rare selenocysteine amino acid, which is important for enzymatic activity. The three deiodinases are membrane homodimers, and the active site is oriented towards the cytosol [53]. More specifically, D1 and D3 are located in the plasma membrane, whereas D2 is in the endoplasmic reticulum membrane [54]. D1 and D2 both catalyse the 5′-phenolic ring deiodination of T4 and, thus, its conversion to T3. The two enzymes have different efficiency since D2 Km for T4 is in the nanomolar range, whereas D1 is in the micromolar range [5,55,56]. Furthermore, D1 has a longer half-life (12 h) than D2 (20–30 min) [57,58] and, in normal conditions, T4 is a better substrate for D2 than for D1. However, D1 (but not D2) has a dual specificity since it also allows the 5-tyrosyl ring deiodination of THs, thus acting as a scavenger enzyme in peripheral tissues to deiodinate iodothyronines (including sulphated iodothyronines) as well as other derivatives, clearing these compounds from circulation and recycling freed iodine [59]. About 20% of circulating T3 in humans (and approximately 50% in rats) is secreted by the thyroid, either through direct synthesis or by T4 to T3 deiodination by D1 and D2, while peripheral T4 to T3 conversion accounts for the remaining 80% [58]. The fact that in euthyroid patients the administration of propylthiouracil (PTU), a D1-selective inhibitor, reduces plasma levels of T3 by 20–30%, while in hyperthyroid patients the estimated reduction of plasma T3 is about 50%, supports the conclusion that D1 is the major responsible for circulating T3 increase in hyperthyroidism [60]. D2 is considered the deiodinase mainly responsible for local production of T3 in the tissues, and its activity increases in hypothyroid conditions, whereas in hyperthyroidism it is inactivated by selective ubiquination [61]. D1 and D2 are both important for TH homeostasis; however, data obtained from double Dio1 and Dio2 KO mice showed that these deiodinases were not essential for maintaining plasma T3 within the normal range, since compensatory mechanisms such as TSH-induced secretion of T3 by the thyroid and altered clearance of iodothyronines can be activated [59]. The brain relies on D2 activity for T3 availability; in fact, about 80% of brain T3 is produced locally in glial cells lining the third ventricle (astrocytes and tanycytes), where T4 is taken up after crossing the BBB and converted to the active hormone T3 that is distributed via Mct8 to the neighboring neurons where it exerts its regulatory action [62,63,64,65,66]. In tanycytes, more than astrocytes, D2 is believed to play an important role in T3 supply to nuclei of the hypothalamus and, thus, in feedback regulation on TRH-expressing neurons [67]. After selective inactivation of Dio2 in astrocytes (Astro-D2KO mouse model), in fact, the HPT axis is preserved due to D2 activity present in tanycytes [62]. Nonetheless, in Astro-D2KO mice, important alterations in T3-responsive genes were observed, leading to altered expression of specific gene sets in the hippocampus, and consequent mood and behavioural disorders [64].

D2 activity in the brain is regulated by the protein ubiquination/deubiquination process: an excessive T4 induces the ubiquination of D2 and the reduction in its enzymatic activity, whereas the opposite occurs in the case of low levels of T4, D2 is deubiquinated and its activity increases. By this mechanism, when serum T4 is reduced, T3 levels in the tissues are still preserved [61].

The third deiodinase, D3, is the enzyme for the physiological catabolism of thyroid hormone activity. It catalyses only the deiodination of the 5-tyrosyl ring position of T4 and T3 into reverse T3 (rT3) and 3,3′-T2, respectively, both of which have low affinity for TH nuclear receptors, thus limiting the excess of T4 in tissues and maintaining an adequate T3 availability [5]. D3 has a half-life of about 12 h and a study of its crystal structure confirmed that the catalytic domain is oriented towards the cytosol [68]. In normal conditions, D3 is preferentially expressed in embryonic tissues, in the brain, and in the placenta [69]; however, in many pathological conditions, D3 is also induced in several other tissues [69,70]. Dio3, the gene coding for D3, undergoes genomic imprinting which results in a high degree of monoallelic gene expression, thus exposing the subject to a higher risk of developing pathologies [71]. In humans, derangements in genomic imprinting cause important neurological and metabolic syndromes [72,73]. Dio3 is part of a cluster including the delta-like homolog 1 (Dlk1) gene and the Dio3 gene, and in the mouse fetus, its expression is preferentially paternally inherited [74]. However, it is believed that allelic contribution to Dio3 expression follows a tissue/cell-specific pattern and that Dio3 imprinting in the brain is variable among different regions; however, further studies need to be done to address these important regulatory aspects [75].

## 6. THs and Experimental Models of Brain Injury

THs are essential for normal brain development, growth, and neural development and are believed to have a central role in the recovery process after brain injury caused by inflammation, metabolic derangements, or neural death [76].

In recent decades, both in vitro and in vivo studies permitted to investigate in depth TH mechanisms of action, and to highlight TH involvement in metabolic and cognitive recovery after brain injury [77].

In vitro models are fundamental to map the pathways activated in response to neuronal injury and to define TH’s role in neuronal protection and recovery. In a human neuroblastoma cell model, under hypoxic conditions, T3 treatment significantly induces hypoxia-responsive factor 2α (HIF-2α) and different genes important for neural survival and neurogenesis, which would protect cells from injury and induce regeneration [78]. Moreover, in rat brain-derived endothelial cells, it was observed that THs have direct proangiogenic activity, stimulating cell expansion and the formation of tube-like structures through the regulation of vascular endothelial growth factor A (Vegfa) and basic fibroblast growth factor (Fgf-2). At the same time, TH treatment inhibits apoptosis through an increase in antiapoptotic gene B-cell lymphoma 2 (Bcl2) gene expression and a reduction in the expression of proapoptotic gene “Bcl2 associated agonist of cell death” (Bad) [79].

The role of the thyroid hormone in the brain in vivo has been inferred principally from phenotypic evaluation of hypothyroid and hyperthyroid animals, and mainly the rat. Both natural animal models, selected with spontaneous gene mutations mimicking human pathological situations, and transgenic animal models, in which specific genes have been eliminated, made silent, or overexpressed through homologous recombination, have been largely employed to study the effects of THs on the development of the nervous system, brain damage protection, and behaviour modulation.

The evaluation of the effects of pharmacological treatments on brain damage due to various pathologies is typically carried out with wild type animals in which a traumatic brain injury mimicking the brain trauma under study is experimentally induced [80]. Such lesions can be mild, moderate, or severe and can be extensive or localized to specific areas. In studies aiming to evaluate the effects of neuroprotective therapies, focal lesions in the areas of interest are normally induced by interrupting blood supply, thus mimicking an ischemic stroke [80,81] and leading to tissue damage, haemorrhage, edema, and hypoxia, followed by neuronal necrosis and apoptosis [78,82]. After injury, however, some neuronal survival and neurogenic signals emerge, and cells migrates towards the injured cortex where latent progenitor cells are activated and may participate in brain repair and functional recovery [83,84].

Several studies in animal models undergoing controlled cortical injury showed beneficial effects after TH treatment, with the reduction of ischemia and edema and the recovery of brain size [77,78,85]. It is not yet clear how TH treatment reduces edema and it was hypothesized that it may be associated to a regulation of aquaporin proteins and intracranial pressure [85].

In controlled cortical injury, the trauma affects the various regions of the brain differently, the cortex being the most affected and, meanwhile, the most responsive to T4 treatment. Otherwise, the hippocampus and cerebellum, both directly affected by THs under normal conditions, respond modestly to T4 after injury [78]. In the brain, cell death following traumatic or ischemic insults in vivo results in the alterations of anti- and pro-cell death signaling pathways, with consequent activation of death-inducing cysteine proteases and caspases [86,87]. In such a pathological context, the treatment with THs promotes neuronal recovery by inducing the expression of BCL-2 and neurogenesis-related genes and reducing pro-apoptotic BAX expression [78]. Early treatment is critical to prevent irreversible functional impairment of the brain and administration of THs is also extremely advantageous because they are stable, not expensive, and easily storable compounds. Further studies are needed to establish the optimal timing and dosage of THs to achieve the most adequate response.

## 7. THs and In Vivo Models of Cognitive/Emotional Impairment

Animal models have also been used to study the involvement of the THs in modulating behaviour, especially as regards cognitive and emotional aspects. It is known that TH disorders, such as hypothyroidism, can be associated with anxiety and depression [88], and both hypothyroidism and hyperthyroidism are considered risk factors for cognitive impairment, as they can influence its dramatic progression [89]. In KO mice, it was observed that a mutation in the gene for TRα1 induces a 10-fold reduction in the affinity for THs, resulting in behavioural disorders, such as reduced exploratory capacity, locomotor dysfunctions and memory deficits, as evaluated through the open-field test and the novel object recognition task [90]. Similarly, mice expressing a mutation of the TRα1 receptor showed a reduced locomotor activity, measured by the beam walk test, indicating that THs are important for proper development of the cerebellum [91]. Experimentally induced imbalances in TH concentrations in adult mouse models have been shown to produce spatial memory deficits that can be evaluated through behavioural tests such as the open-field, elevated plus-maze, radial arm water maze, and passive avoidance test. These disorders can be compensated by pharmacological treatments such as the administration of T3 to mice with hypothyroidism induced by administration of propylthiouracil (PTU), an inhibitor of Dio1 [92]. Also, some natural products such as the antioxidant flavonoid chrysin (5,7-dihydroxyflavone) were tested on mice in which hypothyroidism has been induced by exposure to methimazole, and cognitive deficits compensation was appreciable [93]. Furthermore, T4 administration significantly improved the ability of euthyroid rats to learn spatial memory tasks and these improvements resulted in being strongly associated with a significant increase in cholinergic activity in the frontal cortex and hippocampus of treated animals, demonstrating a relationship between T4 and cholinergic activity [94].

Furthermore, thyroidectomized adult rats showed a significant worsening of cognitive performances, involving both short and long-term memory, evaluated through the radial arm water maze test, but cognitive impairment was markedly reduced after T4 treatment [95]. If the production of THs is blocked in the first weeks of life in pups through administration of PTU, memory deficits are observed in the maze test [96]. These cognitive deficiencies are more relevant if PTU is administered during gestation and breastfeeding, when lowered TH concentrations can compromise hippocampal neuroplasticity and lead to irreversible neurological, learning, and memory damage in adult rats, easily highlighted through the Morris water maze test [97]. In some studies, the role of physical exercise was also evaluated, and the resulting spatial memory deficits improved in hypothyroid rats after exercise on a treadmill [98,99]. In addition, adult-onset hypothyroid mice have reduced performances in odour discrimination [100], and the offspring of rat mothers deprived of THs during gestation have adult learning and memory deficits in the radial maze test and reduced hearing abilities [101].

Aside from cognitive abilities, emotional behaviours such as fear and anxiety are also modified by TH deficiencies, as assessed using elevated plus-maze and open-field tests [102]. With respect to euthyroid mice, hypothyroid animals showed a greater anxiety, assessed with the open-field test, along with worse spatial orientation, assessed with the water Morris maze test, and reduced motor activity in the forced swim test [103]. Similarly, hypothyroid adult rodents showed greater emotional vulnerability, displaying more frequent anxious behaviours and a state of fear, as evaluated with the open-field and elevated plus-maze tests [102,104,105].

Finally, while most research has been conducted on males, some studies, focused on methimazole-induced gestational hypothyroidism, have also considered the effect of thyroid hormones on females. A recent study demonstrated that gestational hypothyroidism altered some of the dams’ behavioural patterns displayed postpartum, and highlighted that adult male offspring exhibited an altered state of fear, as measured through fear conditioning, whereas female offspring did not [106]. No sex differences had previously been observed in the state of anxiety [107] and fear conditioning [108] of male and female offspring of mothers with gestational hypothyroidism.

## 8. THs and Alzheimer’s Disease

In recent years, thyroid function has gained great attention in the study of the cause-effect relation with Alzheimer’s disease (AD), a chronic neurodegenerative condition and the most common form of dementia. The exact mechanisms of AD are not yet completely understood, even though most studies indicate high levels of beta-amyloid peptide (Aβ), neurofibrillary tangles (NFT), and phosphorylated tau protein, as major triggers to the start and/or progression of the disease [109]. In normal subjects, Aβ is excised from Aβ precursor protein (APP) by β- and γ-secretase and released outside the cell, where it is rapidly degraded. In elderly subjects or in certain pathological conditions, Aβ is less efficiently removed and accumulates, forming Aβ amyloid fibrils which develop into senile plaques, considered among the major causes of neurodegeneration [109].

Population-based studies suggested the existence of a close relationship between TH level alterations and the risk of AD. Hypothyroidism becomes more prevalent with age and untreated subjects have marked learning and memory impairments [110]. Contrasting data have been reported on the effectiveness of TH replacement therapy in hypothyroid patients and, despite normal serum TH levels, some symptoms of cognitive dysfunction may be still present. In fact, serum TH levels do not always reflect the hormonal status in the target tissues, where TH content is also under the control of hormone transporters and deiodinases [63]. In particular, the hippocampus is considered a structure of the brain highly sensitive to THs, and recent studies indicated that the deficit in the hippocampus of hypothyroid adult rats is associated with spatial learning and memory impairment, reduction of brain and hippocampal volume, increased levels of Aβ peptide, excessive Tau protein phosphorylation, and production of pro-inflammatory cytokines [92,111,112]. In such a hypothyroid model, the administration of T3 is reported to markedly improve biological functions and cognitive abilities, thus emphasizing the possible therapeutic relevance of T3 supplementation in the definition of new approaches in AD prevention and cure.

The precise mechanisms of interference of THs in the development of AD are still far from being completely clarified and the data obtained from in vitro studies on murine and human neuroblastoma cell lines and primary cultures of rat cortical neurons have demonstrated that THs have an inhibitory effect on APP gene expression and processing [113,114]. Repression of APP gene expression after T3 exposure requires the interaction of the hormone with the TRs, thus, the liganded receptor interacts with negative TRE present in the first exon of the APP gene. This finding is supported by the reduced TRα gene expression in the hippocampus of AD patients [113,115] and by the increased APP mRNA and protein in DKO mice, deprived of both TRα1 and TRβ isoforms [116].

## 9. Conclusions

Data reported in the literature strongly indicate that THs are essential for normal brain development and function and that they may be relevant factors in the processes driving the recovery from neurological impairment or brain injury. These beneficial effects have been supported by several studies on animal models in which the administration of THs (T4, T3, or the combination of the two hormones) may induce specific cellular pathways, promoting neuronal survival and ameliorating neurological outcomes. However, more research in this field needs to be done to define the appropriate dosage and the most correct timing for optimal TH therapy.

## Figures and Tables

**Figure 1 biomolecules-14-00198-f001:**
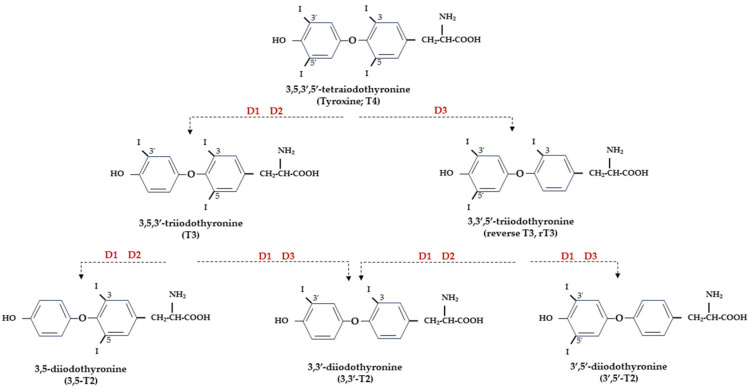
Representation of THs and derived iodothyronines. Deiodinases involved in each type of reaction are indicated as D1, D2, and D3.

**Figure 2 biomolecules-14-00198-f002:**
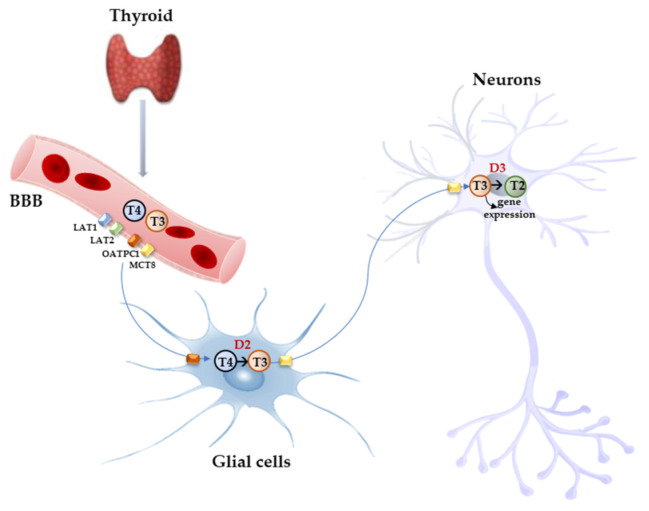
Model proposed for TH signaling in the brain. THs produced by the thyroid cross the BBB through TH transporters (OATP1C1, MCT8, LAT1, and LAT2) and enter astrocytes and tanycytes, where T4 is converted into active T3 by D2 and T3, in turn, with the mediation of MCT8 transporter reaches the adjacent neurons where it regulates the expression of target genes. In the neurons, T3 is deiodinated into 3,3′-T2 by D3 enzyme.

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
