# Peer review of "Factors and Mechanisms of Thyroid Hormone Activity in the Brain: Possible Role in Recovery and Protection"

_biomolecules, 2024, doi:10.3390/biom14020198_

Round 1
Reviewer 1 Report
Comments and Suggestions for Authors
This is a short review of the mechanisms of TH-mediated neuroprotection. The review is well-written, focusing mainly on the transporters, receptors and enzymes that mediate TH entry into the brain, activation, inactivation, and downstream effectors.
This is a decent, if somewhat surface level, overview of the topic. The main criticism that I have is that the authors do not go into much detail of the downstream pathways of neuroprotection. There is brief mention in the head injury section of genes potentially involved, but not in other models or diseases. Most of the focus is on what happens when TH transporters and enzymes are knocked out in rodent models. That is fine, but not necessarily what the title implies.
Comments on the Quality of English LanguageThere are very minor grammatical errors that don't detract from understanding the writing, which is otherwise clear.
Author Response
The authors would like to thank the Reviewer for his comments.
Based on what the Reviewer observed, the title of the ms. has been modified to “Factors and mechanisms of thyroid hormone activity in the brain: possible role in recovery and protection”, to be more precisely in accordance with the content of the ms. Also, some minor corrections were made after the English review (in red in the text, please see the attachment).

Reviewer 2 Report
Comments and Suggestions for Authors
The paper titled "Factors and Mechanisms of Thyroid Hormone Neuroprotection" is a review article that discusses the essential role of thyroid hormones (THs) in brain development, cognitive and emotional functions, and their potential use as a recovery strategy from neuropathological conditions. The authors evaluate the main factors and mechanisms associated with THs in the normal and damaged brain, describe the most common animal models used to study the role of THs in brain damage and cognitive impairment, and highlight the link between TH alterations and increased risk of Alzheimer's disease. The paper concludes that THs are crucial for normal brain development and function and may be relevant factors in the processes driving the recovery from neurological impairment or brain injury. However, more research is needed to define the appropriate dosage and timing for optimal TH therapy. It is a well-written paper, very organized, and very interesting.
A minor comment: line 44 please be clear about T3 and T4 regarding genomic and non-genomics mechanisms.
Author Response
The authors would like to thank the Reviewer for his comments. As requested, a clearer description of the role of T3 and T4 was made by talking about genomic and non-genomic actions (in red in the text, please see the attachment).
"TH action is generally defined as “genomic” when it directly involves the interaction of the biologically active T3 with specific nuclear thyroid hormone receptors (TRs), and “nongenomic” when TH action does not involve nuclear TRs but requires the interaction of T4 with plasma membrane integrin avβ3 receptor, which activates specific intracellular signaling pathways [13,14]."
Reviewer 3 Report
Comments and Suggestions for Authors
Dr. Laura et al. review the factors and mechanisms of neuroprotection by thyroid hormones. They show thyroid hormones have been implicated in neuroprotection from all directions. There are some suggestions that would improve the quality of the manuscript.
1. They state in the abstract and preface that this paper will discuss the important role of thyroid hormone (TH) in normal brain development and function. In fact, however, there is little mention in the text of the association between TH and brain development and function, or the progress of research on this topic. It is understandable that their focus is on neuroprotection, but how TH acts on normal neurogenesis, differentiation migration, etc. is fundamental. It would be easier for the reader to understand if the section "THs and normal brain development" were preceded.
2. The introduction states, 'The purpose of this review is to discuss the major factors and mechanisms that regulate the action of TH in cells'. However, this article reviews almost exclusively in vivo, mouse models. If possible, we would like to see a review of in vitro models for "5. TH and brain injury”, "7. TH and Alzheimer's disease" and the mechanisms of TH involvement in each disease currently being studied.
3. Regarding 6. 'THs and in vivo models of cognitive/emotional impairment' section, I know that TRα1 is the most common in this field, but how advanced is the research on TRβ? Since most human patients with TR dysfunction have TRβ mutations, we would like to see some of the latest findings on in vitro and in vivo models of TRβ.
4. From line 299, the authors state that adult rodents with hypothyroidism showed more emotional vulnerability, more frequent anxiety behaviors and fear states [98,100,101]. In the paper they reference, anxiety is observed in male mice, but have similar studies been conducted in female mice, and is there a sex difference in this study?
5. In Fig. 2, it is preferable to add a figure showing that LAT1 and LAT2 are expressed.
Author Response
Thank you for taking the time to review the manuscript. You will find detailed responses below, with the corresponding revisions (in bold). In the re-submitted version of the manuscript, all corrections in the text are in red (Please see the attached file).
Dr. Laura et al. review the factors and mechanisms of neuroprotection by thyroid hormones. They show thyroid hormones have been implicated in neuroprotection from all directions. There are some suggestions that would improve the quality of the manuscript.
1. They state in the abstract and preface that this paper will discuss the important role of thyroid hormone (TH) in normal brain development and function. In fact, however, there is little mention in the text of the association between TH and brain development and function, or the progress of research on this topic. It is understandable that their focus is on neuroprotection, but how TH acts on normal neurogenesis, differentiation migration, etc. is fundamental. It would be easier for the reader to understand if the section "THs and normal brain development" were preceded.
We thank the Reviewer for the comment, and we added a new section (section 2) entitled “THs and normal brain development” as suggested. Lines 56-77:
“2. THs and normal brain development
THs are essential for normal brain development as they regulate differentiation, migration, maturation, signaling and metabolism of neurons. The function of THs is to provide the right temporal signal for the different stages of brain development to proceed, and the integration of different neuronal systems and association of glia cells to occur [15]. An incorrect synchronism in such developmental processes, would inevitably lead to permanent deleterious outcome. TH action is particularly critical starting from the late fetal stages until 3-4 week of postnatal period in rodents [16], and the first postnatal month in humans [17]. For these reasons, most research on the role of THs in brain development has concentrated in the perinatal and early postnatal period and both the absence or the excess of THs in this critical time window can cause significant and irreversible structural and functional damage, with a relevant impact on the efficiency of the neurotransmitter system [18]. The presence of TH signaling components at very early stages of brain development, before the onset of fetal thyroid function, points to maternal THs supply as determining factors for proper fetal brain development in the early stages [19]. Interestingly, human data indicate that maternal contribution to the fetus, even though in a smaller percentage, fulfils an important protective role until birth [17]. In fact, children born to mothers with thyroid disease experience an increased risk of neurologic and psychiatric diseases later in life [20,21]. Many of these problems are the expression of impaired TH signaling at different component levels, such as TH transporters, deiodinases, and TRs, but also reflect the alteration in the development of neurotransmitters in the central nervous system [22].”
2. The introduction states, 'The purpose of this review is to discuss the major factors and mechanisms that regulate the action of TH in cells'. However, this article reviews almost exclusively in vivo, mouse models. If possible, we would like to see a review of in vitro models for "5. TH and brain injury”, "7. TH and Alzheimer's disease" and the mechanisms of TH involvement in each disease currently being studied.
Thank you for the comment. According with the Reviewer’s suggestions, we added a new paragraph in section 6 (section 5 of the previous version), and we changed the title of section 6 in “THs and experimental models of brain injury”. Lines 246-256:
“In vitro models are fundamental to map the pathways activated in response to neuronal injury and to define TH role in neuronal protection and recovery. In a human neuroblastoma cell model, under hypoxic conditions, T3 treatment significantly induces hypoxia-responsive factor 2a (HIF-2a) and different genes important for neural survival and neurogenesis, which would protect cells from injury and induce regeneration [83]. Moreover, in rat brain-derived endothelial cells it was observed that THs have direct proangiogenic activity, stimulating cell expansion and formation of tube-like structures through the regulation of vascular endothelial growth factor A (Vegfa) and basic fibroblast growth factor (Fgf-2). At the same time, TH treatment inhibits apoptosis through increase of antiapoptotic gene B-cell lymphoma 2 (Bcl2) gene expression and reduction of expression of proapoptotic gene Bcl2 associated agonist of cell death (Bad) [84].”
A second paragraph was added in section 8 (section 7 of the previous version). The added paragraphs are in red in the revised text. Lines 375-383: “The precise mechanisms of interference of THs in the development of AD are still far to being completely clarified and the data obtained from in vitro studies on murine and human neuroblastoma cell lines and primary cultures of rat cortical neurons have demonstrated that THs have an inhibitory effect on APP gene expression and processing [121,122]. Repression of APP gene expression after T3 exposure requires the interaction of the hormone with the TRs, thus, the liganded receptor interacts with negative TRE present in the first exon of APP gene. This finding is supported by the reduced TRa gene expression in hippocampus of AD patients [121,123] and by the increased APP mRNA and protein in DKO mice, deprived of both TRa1 and TRb isoforms [124].”
3. Regarding 6. 'THs and in vivo models of cognitive/emotional impairment' section, I know that TRα1 is the most common in this field, but how advanced is the research on TRβ? Since most human patients with TR dysfunction have TRβ mutations, we would like to see some of the latest findings on in vitro and in vivo models of TRβ.
To our knowledge, animal models of cognitive/emotional impairment focus only on the TRα1 isoform, which is believed to have the major role in brain development and function, accounting for 70-80% of total TR expression in the brain (Schwartz, H.L.; Strait, K.A.; Ling, N.C. Oppenheimer JH. Quantitation of rat tissue thyroid hormone binding receptor isoforms by immunoprecipitation of nuclear triiodothyronine binding capacity. J. Biol. Chem. 1992, 267, 11794-9).
4. From line 299, the authors state that adult rodents with hypothyroidism showed more emotional vulnerability, more frequent anxiety behaviors and fear states [98,100,101]. In the paper they reference, anxiety is observed in male mice, but have similar studies been conducted in female mice, and is there a sex difference in this study?
Few studies have looked at females, however there are studies on gestational hypothyroidism where both male and female offspring have been studied. We then inserted a new paragraph at the end of section 7 (section 6 of the previous version). Lines 340-347: “Finally, while most research has been conducted on males, some studies, focused on methimazole-induced gestational hypothyroidism, have also considered the effect of thyroid hormones on females. A recent study demonstrated that gestational hypothyroidism altered some of the dams’ behavioral patterns displayed postpartum, and highlighted that adult male offspring exhibited an altered state of fear, as measured through fear conditioning, whereas female offspring did not [112]. No sex differences had previously been observed in the state of anxiety [113] and fear conditioning [114] of male and female off-spring of mothers with gestational hypothyroidism.”
5. In Fig. 2, it is preferable to add a figure showing that LAT1 and LAT2 are expressed.
As suggested by the Reviewer, we added LAT1 and LAT2 in Fig.2.

Round 2
Reviewer 3 Report
Comments and Suggestions for Authors
I am very pleased with the author's sincere and thoughtful response to the reviewer and to the reader. I especially appreciate the detailed description of the effects of thyroid hormones on normal brain development in section 2, as requested.